# Optimising an Infusion Protocol Containing Cefepime to Limit Particulate Load to Newborns in a Neonatal Intensive Care Unit

**DOI:** 10.3390/pharmaceutics13030351

**Published:** 2021-03-08

**Authors:** Anthony Martin Mena, Morgane Masse, Laura Négrier, Thu Huong Nguyen, Bruno Ladam, Laurent Storme, Christine Barthélémy, Pascal Odou, Stéphanie Genay, Bertrand Décaudin

**Affiliations:** 1Groupe de Recherche sur les Formes Injectables et les Technologies Associées, University Lille, CHU Lille, ULR 7365—GRITA, F-59000 Lille, France; morgane.masse@univ-lille.fr (M.M.); laura.negrier.etu@univ-lille.fr (L.N.); huong.hup8993@gmail.com (T.H.N.); ladam.bruno@gmail.com (B.L.); christine.barthelemy@univ-lille.fr (C.B.); pascal.odou@univ-lille.fr (P.O.); stephanie.genay@univ-lille.fr (S.G.); bertrand.decaudin@univ-lille.fr (B.D.); 2Évaluation des Technologies de Santeé et des Pratiques Meédicales, University Lille, CHU Lille, ULR 2694—METRICS, F-59000 Lille, France; laurent.storme@chru-lille.fr

**Keywords:** cefepime, particulate matter, NICU, Infusion protocol, filtration, drug reconstitution, infusion, intravenous

## Abstract

Background: In neonatal intensive care units (NICUs), the simultaneous administration of drugs requires complex infusion methods. Such practices can increase the risk of drug incompatibilities resulting in the formation of a particulate load with possible clinical consequences. Methods: This paper evaluates strategies to reduce the particulate load of a protocol commonly used in NICUs with a potential medical incompatibility (vancomycin/cefepime combination). The protocol was reproduced in the laboratory and the infusion line directly connected to a dynamic particle counter to evaluate the particulate matter administered during infusion. A spectrophotometry UV assay of cefepime evaluated the impact of filters on the concentration of cefepime administered. Results: A significant difference was observed between the two infusion line configurations used in the NICU, with higher particulate load for cefepime infused via the emergency route. There was no change in particulate load in the absence of vancomycin. A filter on the emergency route significantly reduced this load without decreasing the cefepime concentration infused. Preparation of cefepime seemed to be a critical issue in the protocol as the solution initially contained a high level of particles. Conclusion: This study demonstrated the impact of a reconstitution method, drug dilution and choice of infusion line configuration on particulate load.

## 1. Introduction

Intravenous (IV) infusion is a common practice in neonatal intensive care units. However, one of the main difficulties encountered is the limited number of venous accesses for the number of drugs to be injected. As a result, infusion devices are becoming more and more complex to ensure the simultaneous administration of drugs that are sometimes incompatible with each other and can have a clinical impact [1,2] with complications such as phlebitis, pulmonary or renal embolisms [3,4,5] and inflammations [6,7]. Some of these can be fatal [8,9]. In neonatal care units, the obligation to administer only small volumes is also present, which may mean that sometimes drugs are not completely dissolved. The concentrations are different from those used in adults and low infusion rates have to be applied, thus increasing the risk of physical and chemical incompatibilities between drugs [10].

In paediatric and neonatal intensive care units (PICUs and NICUs), data on drug combination incompatibilities is often incomplete [11,12]. Physical and chemical incompatibilities, visible or not, can lead to a loss of active ingredient [13], result in colour change, pH modification, opalescence, gas release or precipitate formation. A precipitate consists of particles that are visible and/or not visible to the naked eye, most commonly from an acid-base reaction. A particulate load may also be present in reconstituted or ready-to-use drugs. The European Pharmacopoeia (EP) dictates thresholds for static particle counting of sizes ≥10 µm and ≥25 µm [14].

To prevent the administration of these particles, optimised medical devices (multi-lumen extension sets) can be used to reduce particulate load [15,16,17] and therefore the risk of incompatibilities [18,19]. However, Bradley et al. have reported that the use of a multi-lumen device does not prevent the formation of precipitate at the end of intravenous tubing [5]. Another solution is an in-line filtration to limit the administration of particles, particularly in neonatal and paediatric units [20,21,22], reducing associated respiratory, haematological and renal complications [7]. Whereas in some studies, the use of in-line filters in PICUs and NICUs has led to a significant decrease in the occurrence of systemic inflammatory response syndrome (SIRS), thrombosis and sepsis [6,21], other studies point to an absence of data supporting the significance of filters in reducing sepsis [23], phlebitis [24] or in preventing mortality or morbidity in newborns [25,26].

The combination of vancomycin and cefepime is noted by some authors to be a possible cause of drug incompatibility [27,28,29]. However, data is contradictory depending on operating conditions [29,30,31]. Observation of practices in the NICU at the Lille University Hospital has shown that these drugs are frequently combined to treat newborns. The objective of this work was to estimate the particulate load induced by the joint infusion of these two drugs under normal clinical conditions and to propose solutions to limit its administration to patients.

## 2. Materials and Methods

### 2.1. Experiments, Devices and Drugs

The infusion conditions studied (concentrations, infusion rates) are those applied in the NICU with the use of a single-lumen catheter (epicutaneo-cava catheter, 2184.00, Vygon, Ecouen, France). The two commonly used infusion lines are shown in Figure 1A,B and consist of two 3-way Y-extension lines (PY3101KNCM, Cair, Lisieux, France). The choice of infusion line assembly depends on the nurse. The two configurations differ as far as the so-called emergency unfiltered route is concerned, infusing either cefepime (cefepime dihydrochloride monohydrate, Mylan 1 g, Fretin, France) infusion (CEF infusion line) or caffeine (caffeine citrate, Cooper 25 mg/mL, 2 mL, Melun, France) infusion (CAF infusion line). This emergency route is dedicated to drugs that cannot be filtered or need short infusion times. Drugs that are continuously infused are always filtered. The lipid emulsion is filtered with a 1.2 µm porosity filter (ref: 807.504, Vygon, Ecouen, France); a 0.22 µm porosity filter (ref: 807.204, Vygon, Ecouen, France) is placed downstream of the second channel of the 3-way Y-extension line for binary parenteral nutrition, vancomycin (hydrochloride vancomycin, Sandoz 250 mg, Levallois-Perret, France) and caffeine (CEF infusion line) or cefepime (CAF infusion line) solutions (Figure 1A,B).

The therapeutic infusion protocol is presented in Table 1. Depending on clinical situations, infusions of cefepime and caffeine are simultaneous, two to three times daily, for 30 min followed by rinsing with saline solution (SS) (0.9% NaCl, Baxter, Guyancourt, France) to guarantee total infusion of the medicine. As the infusion rates for cefepime and caffeine are different (5.1 mL/h vs. 3.6 mL/h), the flushing time for these two lines is not identical (Table 1).

The different reconstitution/dilution solutions used were water for injections (WFI, Baxter, Guyancourt, France), saline solution (SS, Baxter, Guyancourt, France) and 5% glucose solution (G5%, Baxter, Guyancourt, France).

The two infusion line configurations were replicated in vitro in the laboratory with two modifications: binary parenteral nutrition solution and lipids were replaced with a 5% glucose solution (G5%) (Baxter, Guyancourt, France), to focus only on potential drug incompatibilities and drug particles since the particle counter is unable to differentiate between lipid globules and particles (Figure 2A,B) and (Table 1).

The protocol adapted to the laboratory is shown in Figure 3 and divided into four periods. Because of the opening hours of the research laboratory, the duration of the experiments was set at 8 h with a single infusion of cefepime and caffeine, representing one of the three identical consecutive cycles of infusion that occur over 24 h.

### 2.2. pH Measurement

pH values were measured directly in the syringes after reconstitution and dilution of vancomycin, cefepime and caffeine with a calibrated pH meter (SB70P Symphony, VWR International, Fontenay-sous-bois, Singapore, France). The pH of the diluents was also determined. During each infusion period, pH was measured over the total volume collected at the infusion line exit (Figure 3). For each solution, three measurements were taken.

### 2.3. Osmolality Measurement

The osmolality of each solution and solution mixture was measured by a micro-sample osmometer (Fiske 210, Advanced Instrument, Norwood, MA, USA). This freezing-point osmometer evaluates osmolality with a 20 µL solution. The three osmolality measurements were taken on the same samples as the pH.

### 2.4. Static Particle Load Analysis

For static particulate load analysis, the various drug solutions were analysed by APSS-2000 (Automated Parenteral Sampling System, Particle Measuring Systems, Dourdan, France). This is a particle counter consisting of an SLS-1000 syringe containing the sample, a Liquilaz E20P light obscuration spectrometer and Sampler Sight-Pharma operating software. This apparatus meets the requirements of the EP. The APSS-2000 can measure the particulate load of a static sample for particle sizes between 1.5 and 125 µm.

Six prepared syringes were analysed per drug; four samples of 6 mL were taken from each syringe. The first sample was rejected. According to EP monograph 2.9.19, the number of particles of 10 and 25 µm or larger should not exceed 6000 and 600 respectively for each syringe. The results were expressed as medians [min; max] for each particle size.

### 2.5. Dynamic Particle Load Analysis

For the analysis of dynamic particle counting, we used the Qicpic dynamic image analysis device (Sympatec GmbH Inc, Clausthal-Zellerfeld, Germany) combined with a Lixell module (Sympatec GmbH, Clausthal-Zellerfeld, Germany). The frame rate was 10 Hz and was synchronised with a high-speed camera that captures up to 500 images at 1024 × 1024 pixels per second. The Qicpic analyser with Windox 5.0 software determined particle sizes between 1 µm and 30 mm and provided dynamic imaging analysis. The external configurations were connected to the external connectors of the Lixell module via luer-lock connectors. In this study, the outlet catheter of the IV administration set was directly connected to the Qicpic to obtain an accurate measurement of the particulate load administered per minute throughout the infusion. Data analysis was presented either as histograms and tables for the number of total particles or as tables for the number of particles ≥2 µm, ≥5 µm, ≥10 µm, ≥15 µm and ≥25 µm. The results also made it possible to define the number of particles as a function of infusion time (Q(t)). The number of particles was observed at different time intervals: T0 to T8H, T0 to T4H, T4H to T5H and T5H to T8H. The cumulative particle distribution (Q(x), expressed as a percentage) was defined for the CEF infusion line.

### 2.6. Evaluation of the Origin of Particles and Proposed Strategy to Reduce Particulate Load

#### 2.6.1. Modalities for Reconstituting/Diluting Cefepime

The summary of product characteristics (SmPC) was respected. It states that cefepime is compatible with the following solvents and solutions: WFI, SS (with or without 5% glucose), 5 or 10% glucose, Ringer lactate solution (with or without 5% glucose), lidocaine hydrochloride solution, 1/6M sodium lactate. Static particle counting (*n* = 6) was performed under the conditions used in the neonatal service (WFI) and with other solvents or reconstitution/dilution solutions: WFI/G5%, SS/SS, G5%/G5%.

#### 2.6.2. Study of Cefepime/Vancomycin Interaction

To determine if particulate load was due to an incompatibility between vancomycin and cefepime, a modification was made to the CEF infusion line. Vancomycin was replaced by its diluent (G5%) (CEF infusion line without vancomycin) (Figure 4A).

#### 2.6.3. Adding an Additional Filter

To assess the impact of particles resulting from the preparation of the diluted cefepime solution, another infusion line model was used adding a 0.22 µm filter on the so-called emergency route (CEF filter infusion line) (Figure 4B). A dose of cefepime solution was infused to observe the impact of the filter on cefepime concentration.

### 2.7. Determination of Cefepime Dihydrochloride Monohydrate Concentration

The calibration range and assays were performed on a UV spectrophotometer (UV2550, Shimadzu, Kyoto, Japan) at 248 nm. The EP cefepime dihydrochloride monohydrate standard reference (Y00007603, EDQM, Strasbourg, France) was used for the calibration range. Validation was completed over three consecutive days by determining the following parameters: specificity, linearity, limits of detection (LOD) and quantification (LOQ). LOD and LOQ were set at a risk β of 5% and the critical level (CL) at a risk α of 5%, so that LOD and LOQ were defined respectively by 2- and 4-times CL.

Cefepime was present as cefepime dihydrochloride monohydrate in drug vials. The only excipient present in medical cefepime was L-arginine, a non-UV absorbing amino-acid.

In order to assess the impact of filters, the concentration of cefepime dihydrochloride monohydrate was compared downstream and upstream of filters at the egress of the 3-way Y-extension line and on the emergency route.

The expected infusion concentrations for the emergency route and the 3-way Y-extension line were 20.5 mg/mL and 16.5 mg/mL respectively. The different samples were diluted 1/500 with 0.9% SS to avoid signal saturation. Six different assays were analysed per condition. Each sample was its own control.

### 2.8. Statistics

Data is presented as medians [minimum; maximum] in the text and graphs. All graphs were plotted and analysed by GraphPad Prism 6 Software (San Diego, United States). Given the size of our samples, two types of non-parametric tests were used: a Mann–Whitney test for unpaired samples and a Wilcoxon test for paired samples.

*p* values < 0.05 were considered statistically significant.

## 3. Results

### 3.1. pH and Osmolality Measurements

The pH and osmolality results for each solution (with or without drug) are presented in Table 2A.

The pH median changes during infusion (Table 2B).

### 3.2. Static Particle Counting

The results of the static count of vancomycin, caffeine and cefepime are presented in Table 3. According to the EP specifications for containers with a volume ≤ 100 mL, the vancomycin and caffeine solutions conformed but the cefepime solution did not.

### 3.3. Dynamic Measurement of Particulate Contamination during Infusion

#### 3.3.1. Comparison of the Particulate Load of CEF and CAF Infusion Lines

A difference in particulate load during the 8-h infusion was observed between the two assembly lines. A peak was visible during the simultaneous infusion of cefepime, caffeine and vancomycin with the CEF infusion line (Figure 5A,B).

The graph of the cumulative distribution of particles on the CEF infusion line showed that 99% of the particles were smaller than 10 µm (Figure 5C).

A significant difference in total particulate load was observed between the CEF infusion line and the CAF infusion line (*p* < 0.01) (Figure 6A). Analysis of particulate load during infusion was then divided into three parts to pinpoint the critical period. The first period, T0 to T4H (Figure 6B), corresponded to the time duration of simultaneous infusion of vancomycin and G5% and preceded the infusion of cefepime and caffeine. The second period, T4H to T5H (Figure 6C), corresponded to the infusion of cefepime and caffeine and rinsing of the extension lines. The last period, T5H to T8H (Figure 6D), corresponded like the first, to a return to vancomycin and G5% only.

Over the period T0 to T4H, no significant difference in particulate load was observed between the CEF and CAF assembly lines (*p* = 0.4848) (Figure 6B) whereas a significant difference was noted during the period T4H to T5H (*p* < 0.01) (Figure 6C) as well as during the third period (*p* < 0.05) (Figure 6D).

The particulate charge as a function of particle size was also studied over the periods T0-T8H and T4H-T5H. The results are presented in the Appendix A.

#### 3.3.2. Comparison of Particulate Load between the CEF Infusion Line with Vancomycin and without

The total particulate load of the CEF infusion line without vancomycin (Figure 4A) was compared to that of the CEF infusion line with vancomycin (Figure 2A). No significant difference in particulate load was observed between them during the total infusion time (*p* = 0.9307) (Figure 7A) nor at periods T0-T4H, T4H-T5H and T5H-T8H (*p* = 0.5887, *p* = 0.8182 and *p* = 0.2403) (Figure 7B–D). The absence of vancomycin did not statistically affect particulate load.

Particulate load as a function of particle size was also studied over the periods T0-T8H and T4H-T5H. The results are presented in Appendix A.

#### 3.3.3. Comparison of Particulate Load between the CEF, CEF with Filter and CAF Infusion Lines

The total particulate load of the CEF infusion line with filter on the emergency route (CEF filter infusion line) (Figure 4B) was compared to those of the CEF and CAF infusion lines (Figure 2A,B).

During the total duration of the infusion, the presence of a filter on the so-called emergency route of the CEF infusion line significantly reduced particulate load (*p* < 0.01) (Figure 8A). The same significant difference was to be noted during the period T4H-T5H (*p* < 0.01) (Figure 8C). Over the same period, particulate load was lower with the filter than with the CAF infusion line (*p* < 0.05) and during the period T5H-T8H the difference was significant between the CEF infusion line without filter and the CEF infusion line with filter (*p* < 0.01), regardless of particle size (Appendix A). However, during the periods T0-T4H (*p* = 0.0649 and *p* = 0.6991) (Figure 8B,D), the presence of the filter made no significant difference.

The particulate load as a function of particle size was also studied over the periods T0-T8H and T4H-T5H and results are presented in Appendix A.

### 3.4. Impact of the Filter on Cefepime Dihydrochloride Monohydrate Concentration

The limits of detection and quantification were 0.309 µg/mL and 0.618 µg/mL respectively.

The presence of filters had no significant effect on the concentration of cefepime dihydrochloride monohydrate (*p* = 0.3125) (Figure 9).

## 4. Discussion

This study demonstrated the impact of drug reconstitution/dilution methods and the choice of infusion line on the particulate load administered to patients. This is the first study to address the particulate load of a cefepime preparation and underlines the advantage of adding a 0.22 µm filter to reduce the number of particles administered to patients.

Currently, no standard recommendations have been issued by the three main Pharmacopoeias (European, American and Japanese) for dynamic particle counting. This work has been carried out in our laboratory using a new technique for the dynamic counting of non-visible particles (<50 µm) which indicates the potential particulate load to which patients are exposed [16,17,22,32,33,34]. With the use of this appliance, different assembly lines used for infusion protocols in intensive care units (adult, paediatric and neonatal) can be reproduced and studied in vitro. The results obtained make it possible to compare the particulate load of two infusion lines and thus highlight the reduced exposure to particles through the use of filters [16,17,22,33].

The particle counts for each drug solution prior to administration revealed a number of particles for the cefepime solution exceeding the threshold recommended by the EP 2.9.19 monograph regardless of the method used. The preparation of cefepime therefore remained a critical point in the protocol due to its high initial level of particulate load. It is therefore necessary to filter this drug during infusion to minimise the administration of particles. As no work has so far targeted this high particulate load, an analysis of the various marketed presentations of cefepime should be considered. Indeed, other publications have already shown variable particulate loads in the three different formulations of another antibiotic (cefotaxime) [35]. It would seem that this higher load is more likely to affect generic drugs [36]. For the cefepime solution, the use of filters is essential: in the absence of incompatibility, they act as a barrier to particulate load while preserving the desired infused cefepime concentration. The vancomycin and caffeine solutions met EP standards.

During the dynamic particle load assessment, an adaptation of the protocol used in the care unit was made for its in vitro transposition. The results of the count clearly showed a difference between the two assembly lines. The CEF infusion line indicated a significant particulate peak during the infusion period of cefepime and caffeine. All results showed that the type of infusion line had a strong influence on particulate load: the CAF infusion line seemed to be more secure than the CEF infusion line.

Despite the use of a 0.22 µm filter, a residual particulate load was constantly observed that did not interfere with the observations made on the peak during the infusion of cefepime. These observations on residual particle load have been well described in our previous articles [22,34].

Tests were performed replacing vancomycin with a blank and placing a filter on the emergency route when cefepime was infused to determine whether its particulate load was primarily due to drug incompatibility or the load already present in reconstituted and diluted drug solutions.

First of all, the replacement of vancomycin by a 5% glucose solution indicated that the particulate load of the CEF infusion line was not the result of incompatibility between cefepime and vancomycin. Some authors suggest an incompatibility between vancomycin (10 mg/mL) and cefepime (200 mg/mL). Our results, at the studied concentrations (vancomycin: 4.1 mg/mL and cefepim: 17.5 mg/mL) and flows, were in line with studies by Raverdy et al. (vancomycin: 10 mg/mL and cefepim: 83.3 mg/mL) and Berti et al. (vancomycin: 3.69–4.46 mg/mL and cefepim: 1.61–2.22 mg/mL) which demonstrated that there is no incompatibility between these two drugs [27,28,29,30,31]. Our data complements already available data with new concentrations and test conditions. Our study focused only on the possible interaction between vancomycin and cefepime. The compatibility of vancomycin with commercial ready-to-use parenteral formulations has been investigated previously [37,38]. Vancomycin has been shown to be compatible with some mixtures (Nutriflex Lipid Special, Kabiven, Nutriflex Omega Special); however, it is not compatible with others (Olimel N9E and Smofkabiven). Cefepime is compatible in a 1:1 ratio with the specific lipid emulsion (Nutriflex Lipid Special) [37]. The compatibility of cefepime and vancomycin with lipid emulsions selected for nutrition should also be checked so as to introduce the proposed procedure into NICUs.

Finally, regardless of the time period, the number of particles dropped significantly after adding a filter on the emergency route. This shows that the majority of particles came directly from the emergency route with cefepime. In addition, during the infusion period of cefepime and caffeine, the particulate rate with the CEF filter infusion line was significantly lower than that with the CAF infusion line. Finally, the impact of cefepime on particulate load extended beyond its infusion period and track rinsing. The particulate load found on the CEF infusion line during the last infusion period (T5H-T8H) was significantly higher than that of the CAF infusion line or the CEF infusion line with filter, indicating that there was a residual release of particles, probably present in the tubing.

All the results clearly showed that cefepime infusion is mainly responsible for the particulate load initially observed in the CEF infusion line. However, as indicated above, in the absence of a comparison of cefepime from different manufacturers, it is impossible to extend our result obtained with one version from one manufacturer (Mylan) to all versions on the market. This reinforces the proposal to standardise the infusion line for this protocol by integrating a filter for drugs like cefepime that risk administering particles. Benlabeb et al. have summarised various studies that highlight the impact of particles on micro-circulation and the importance of placing filters on neonatal intensive care infusion lines [39]. However, since some drugs (coagulation factors, ambisome, fungizone, rifampicin, phenobarbital, insulin) cannot be filtered [40], the emergency route cannot be systematically filtered. This study and that of Masse et al. confirmed that cefepime and vancomycin solutions can be filtered without fear of losing the desired concentration [36]. Despite there being a low level of particles in caffeine solution compared to the two other drugs, it could be useful to check the concentration of caffeine solution after filtration. We are currently pursuing an evaluation work to cover all cefepime formulations available on the French market to assess particulate loads and the impact of filters on the level of cefepime infused. In this respect, the effect of excipients must be taken into account as well as the clinical impact on observed dosage loss [41]. However, the literature is reassuring for the moment. The efficacy of the infusion is the essential issue and may prevail despite dosage loss. This remark has been made many times for other products [41,42,43].

It is necessary to transpose the obtained results into the context of infusion protocols in use in NICUs. Indeed, our results offer an estimation of particulate load that can be infused to newborns over a period of 8H instead of the 24H of the initial protocol in the care unit. Neonates receive two or even three infusions of cefepime depending on the prescription and therefore potentially receive twice (or even three times) the particulate load estimated during our tests. It would be worth noting whether a cumulative effect of particulate load is present during a second or third infusion of cefepime. The efficacy and integrity of the filter also requires evaluation throughout its period of use.

## Figures and Tables

**Figure 1 pharmaceutics-13-00351-f001:**
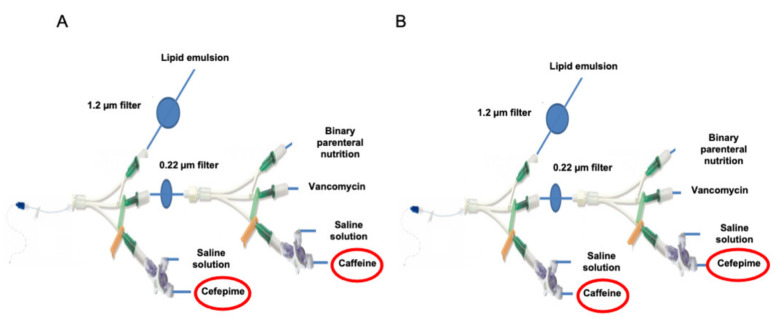
Infusion line used in the neonatal intensive care unit. CEF infusion line with cefepime placed on the unfiltered emergency track (**A**); CAF infusion line with caffeine placed on the unfiltered emergency route (**B**).

**Figure 2 pharmaceutics-13-00351-f002:**
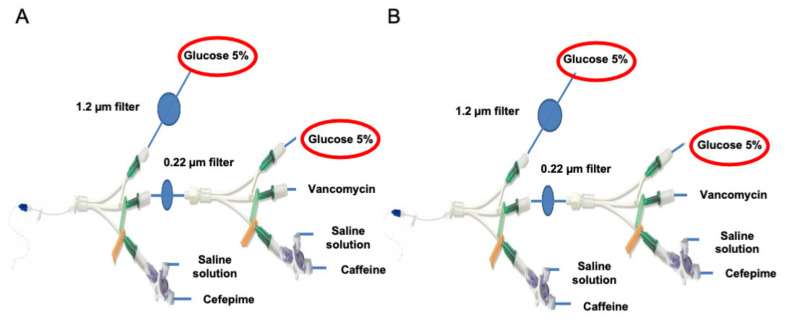
Replicated infusion line used in the laboratory. Modified CEF infusion line with replacement of binary parenteral nutrition infusion and lipid emulsion infusion by G5% infusion (**A**); Modified CAF infusion line with replacement of binary parenteral nutrition infusion and lipid emulsion infusion by G5% infusion (**B**).

**Figure 3 pharmaceutics-13-00351-f003:**
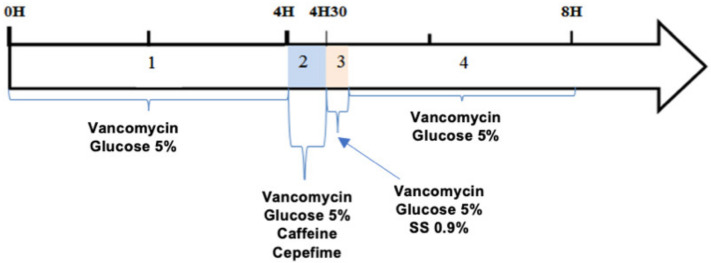
Infusion protocol adapted in vitro to the laboratory. 4 periods: (1) period T0 to T4H: Perfusion of vancomycin and G5%; (2) period T4H to T4H30: Perfusion of vancomycin, cefepime, caffeine and G5%; (3) T4H30-T4H41: Perfusion of vancomycin, G5% and rinsing with saline solution of the cefepime and caffeine routes; (4) period T4H41 to T8H: Perfusion of vancomycin and G5%.

**Figure 4 pharmaceutics-13-00351-f004:**
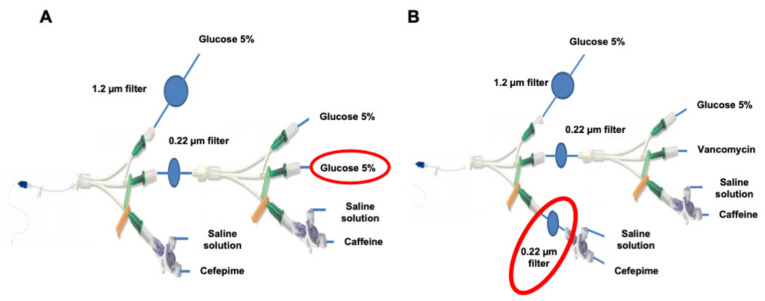
Modified infusion line used in the laboratory. Modified CEF infusion line with replacement of vancomycin infusion by G5% infusion (**A**); Modified CEF infusion line with a 0.22 µm filter positioned on the emergency route (**B**).

**Figure 5 pharmaceutics-13-00351-f005:**
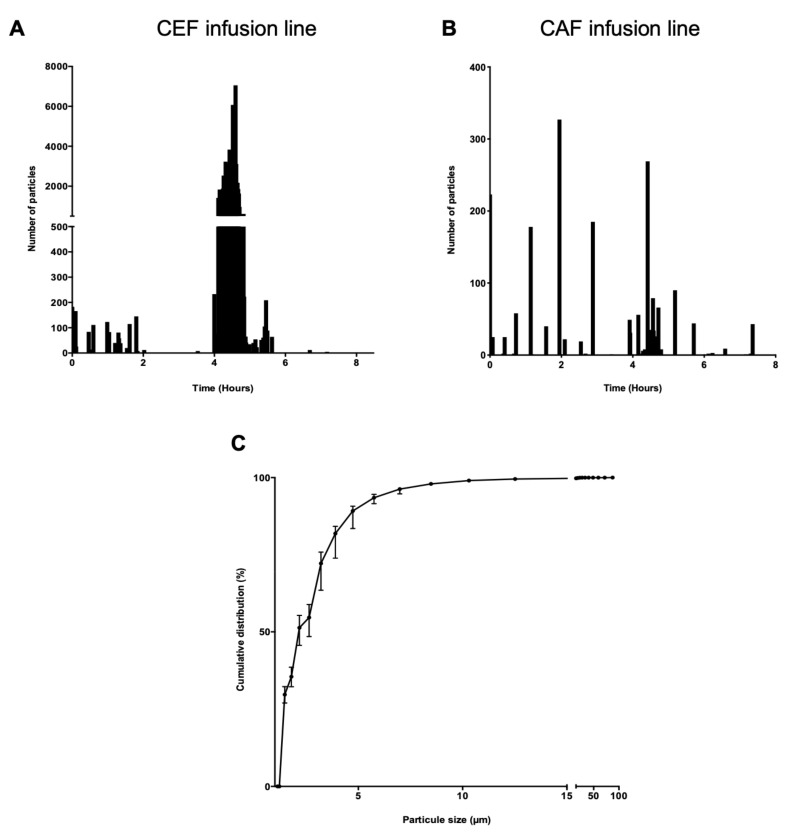
Example of particle distribution over time. CEF infusion line (**A**), CAF infusion line (**B**). Cumulative distribution of particles in a CEF infusion line, (*n* = 6) (**C**).

**Figure 6 pharmaceutics-13-00351-f006:**
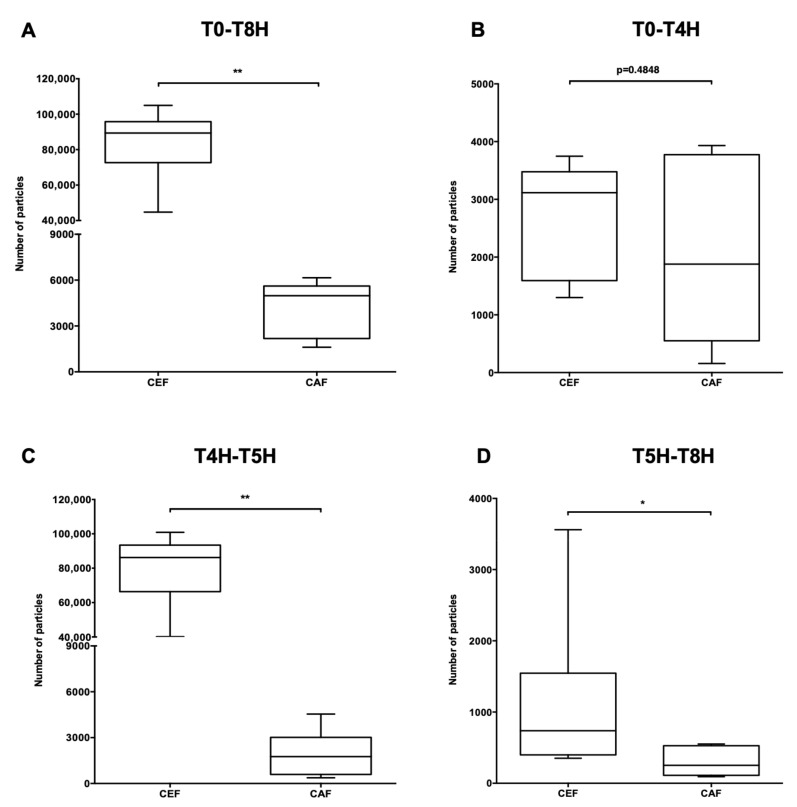
Impact of the choice of infusion line on particulate load. Comparison of total particulate load as a function of infusion line at times T0-T8H (**A**); T0-T4H (**B**); T4H-T5H (**C**); T5H-T8H (**D**). Results are expressed as medians [min; max] (**: *p* < 0.01, *: *p* < 0.05, Mann Whitney, *p* < 0.05, *n* = 6).

**Figure 7 pharmaceutics-13-00351-f007:**
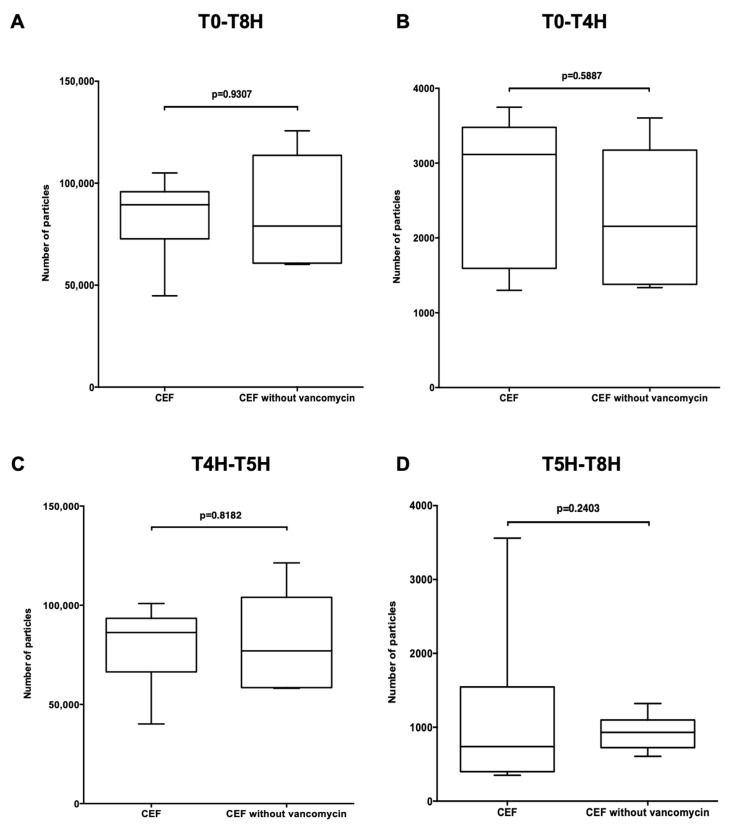
Impact of replacing vancomycin with a G5% infusion on particulate load. Comparison of total particulate load as a function of vancomycin/G5% infusion at times T0-T8H (**A**); T0-T4H (**B**); T4H-T5H (**C**); T5H-T8H (**D**). CEF: infusion line with cefepime on emergency route and with vancomycin; CEF without vancomycin: infusion line with cefepime on emergency route and with replacement of vancomycin by G5%. Results are expressed as medians [min; max] (Mann Whitney, *p* < 0.05, *n* = 6).

**Figure 8 pharmaceutics-13-00351-f008:**
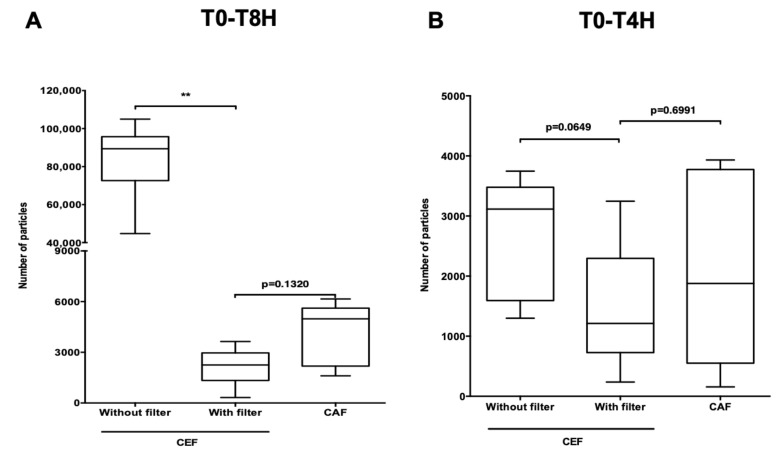
Impact of filter on the emergency route of the CEF infusion line. Comparison of total particulate load between a CEF infusion line without filter and a CEF infusion line with filter and a CAF infusion line at times T0-T8H (**A**); T0-T4H (**B**); T4H-T5H (**C**); T5H-T8H (**D**). CEF without filter: infusion line with unfiltered cefepime on emergency route; CEF with filter: infusion line with filtered cefepime on emergency route; CAF: infusion line with caffeine on emergency route. Results are expressed as medians [min; max] (**: *p* < 0.01, *: *p* < 0.05, Mann Whitney, *p* < 0.05, *n* = 6).

**Figure 9 pharmaceutics-13-00351-f009:**
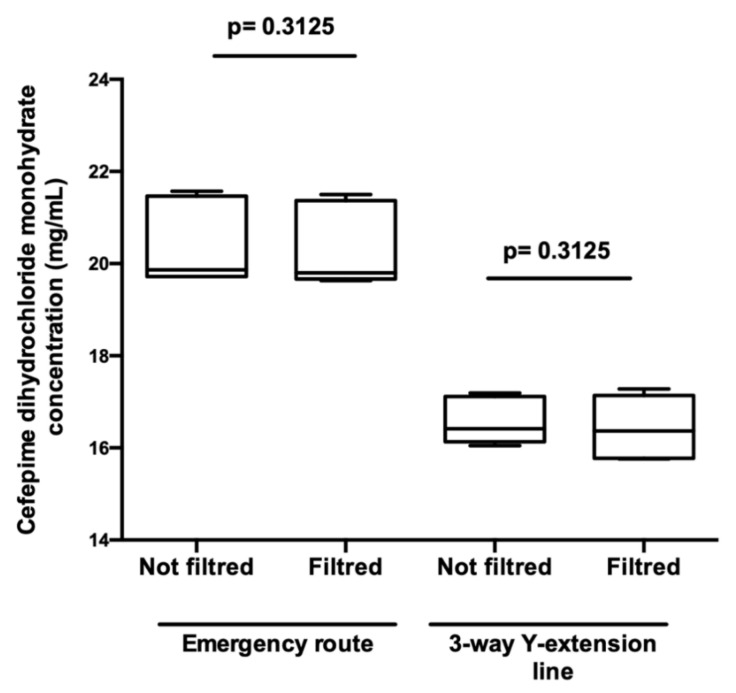
Impact of the filter on cefepime dihydrochloride monohydrate concentration. Comparison of cefepime dihydrochloride monohydrate concentrations between filtered and unfiltered infusion lines: emergency route or 3-way Y-extension line. Results are expressed as medians [min; max] (Wilcoxon, *p* < 0.05, *n* = 6).

**Table 1 pharmaceutics-13-00351-t001:** Reconstitution, dilution, concentration, rate and time of infusion of the different drugs applied in the in vitro protocol (WFI: Water for injections, SS: Saline solution, G5%: 5% Glucose solution).

Medicines	Protocol Used in NICU			In Vitro Protocol	
	Reconstitution/Dilution Solutions	Final Concentration (mg/mL)	Infusion Time in the Unit	Protocol Used In Vitro Compared to Clinical Use	Infusion Time In Vitro
Vancomycin	WFI/G5%	4.1	24 h	No change	8 h
Caffeine	G5%	3.4	30 min every 8 or 12 h	No change	30 min
Cefepime	WFI/SS 0.9%	17.5	30 min every 8 or 12 h	No change	30 min
Binary parenteral nutrition	-	-	24 h	Replaced by G5%	8 h
Lipid emulsion	-	-	24 h	Replaced by G5%	8 h
SS 0.9% 1 (Rinsing of caffeine)	-	-	11 min every 8 or 12 h	No change	11 min
SS 0.9% 2 (Rinsing of cefepime)	-	-	8 min every 8 or 12 h	No change	8 min

**Table 2 pharmaceutics-13-00351-t002:** pH and osmolality measurements of the different infusion solutions (**A**) (*n* = 3). pH and osmolality measurements at the exit of the catheter at different infusion times. T0 to T4H and T5H to T8H: vancomycin and G5% infusion; T4H to T4H30: vancomycin, cefepime, caffeine and G5% infusion; T4H30 to T4H41: vancomycin, G5% and SS infusion (**B**) (*n* = 3).

Table A	Cefepime (WFI/SS)	Vancomycin (WFI/G5%)	Caffeine (G5%)	G5%	SS	
**pH**	4.43 [4.36–4.48]	3.31 [3.25–3.32]	2.42 [2.36–2.47]	3.10 [3.07–3.27]	5.81 [5.79–5.82]	
**Osmolality** **(mOsm/kg)**	380 [380–383]	252 [249–255]	276 [271–298]	293 [292–294]	285 [284–285]	
**Table B**	**Periods T0 to T4H and T4H41 to T8H**		**Period T4H to T4H30**		**Period T4H30 to T4H41**	
**pH**	3.41 [3.39–3.46]		3.52 [3.51–3.52]		4.20 [4.18–4.21]	
**Osmolality** **(mOsm/kg)**	284 [284–291]		318 [317–320]		288 [287–290]	

**Table 3 pharmaceutics-13-00351-t003:** Static count of particles ≥10 and ≥25 µm with the APSS-2000. Median [min–max], *n* = 18.

Medecines (Reconstitution/Dilution Solutions)	≥10 µm	≥25 µm
**Vancomycin (WFI/G5%)**	4250 [3300–5920]	90 [20–230]
**Caffeine (G5%)**	1746 [556–2188]	103 [30–182]
**Cefepime (WFI/SS)**	13,236 [9155–24,239]	595 [156–1326]
**Cefepime (WFI/G5%)**	12,534 [9809–25,620]	1414 [507–2652]
**Cefepime (SS/SS)**	21,782 [5928–34,613]	858 [117–5041]
**Cefepime (G5%/G5%)**	13,638 [6659–23,322]	410 [67–2057]

## Data Availability

Not applicable.

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
