# Peer review of "Optimising an Infusion Protocol Containing Cefepime to Limit Particulate Load to Newborns in a Neonatal Intensive Care Unit"

_pharmaceutics, 2021, doi:10.3390/pharmaceutics13030351_

Round 1

Reviewer 1 Report

The literature suggests an incompatibility between vancomycin and cefepime (200 mg / mL), however the authors explain that based on their results in line with the studies by Raverdy et al. and Berti et al. There is no such incompatibility between these two drugs, in my opinion this vital information in this article should be supported with more information and trials. 

Reviewer 2 Report

The authors of the study: “Optimizing an infusion protocol containing cefepime to limit [the] particulate load to newborns in a neonatal intensive care unit.” prepared and clearly presented the procedures of administering cefepime in one line with caffeine and vancomycin. The analytical studies were well documented and statistically analyzed. The authors did not investigate the drug-fat emulsion interaction (which was replaced with 5% glucose solution) in their model. They wrote: “Our study focused only on the possible interaction between vancomycin and cefepime, but one study has shown that vancomycin may be incompatible with parenteral nutrition [36], for which further trials are therefore necessary”. In article No 36 (Fonzo-Christe C. et al. Nutr. Clin. Métabolisme 31 (1), 24–27, 2017) I cannot find the information cited. 
The compatibility of vancomycin with commercial RTU-type parenteral formulations has been investigated previously [e.g., A, B]. Vancomycin has been shown to be compatible with some mixtures; however, it is not compatible with others (Olimel N9E and Smofkabiven). Cefepime is compatible in a 1: 1 ratio with Nutriflex Lipid Special [A]. It would also be necessary to check the compatibility of cefepime (not only vancomycin) with lipid emulsions selected for nutrition to introduce the proposed procedure in NICUs. The authors’ research explains the problem of the appearance of numbers of particles in the infusion line. The recommendation to add a 0.22 µm filter on the emergency route of a cefepime container is justified. However, it should be emphasized that due to the undocumented compatibility with lipid emulsions, the proposed drugs cannot be administered in the combination presented in Fig. 1A.

A. Bouchoud L. et al. JPEN 37(3), 416-424, 2013.

B. Stawny M. et al., Clin. Nutr. 39(8):2539-2546, 2020.

Reviewer 3 Report

As pointed out in this paper, in the neonatal intensive care unit (NICU), the simultaneous administration of drugs requires complex infusion methods.
This study will provide important information for proper drug administration.

While I think the research design itself is fine, I must point out that the sample size is too small in this study.

It would be easier for the reader to understand if you mention the sample size in the research limitations.

However, as I mentioned in the beginning, this research topic will definitely provide important information in NICUs.
